# Advancing Precision Medicine in South Tyrol, Italy: A Public Health Development Proposal for a Bilingual, Autonomous Province

**DOI:** 10.3390/jpm13060972

**Published:** 2023-06-09

**Authors:** Christian J. Wiedermann

**Affiliations:** 1Institute of General Practice and Public Health, Claudiana—College of Health Professions, 39100 Bolzano, Italy; christian.wiedermann@am-mg.claudiana.bz.it; 2Department of Public Health, Medical Decision Making and Health Technology Assessment, University of Health Sciences, Medical Informatics and Technology, 6060 Hall in Tirol, Austria

**Keywords:** precision medicine, South Tyrol, bilingual population, healthcare workforce, digitalization, data management, biobank, personalized treatment, prevention strategies, pharmacogenomics, healthcare outcomes

## Abstract

This paper presents a comprehensive development plan for advancing precision medicine in the autonomous province of South Tyrol, Italy, a region characterized by its bilingual population and unique healthcare challenges. This study highlights the need to address the shortage of healthcare professionals proficient in language for person-centered medicine, the lag in healthcare sector digitalization, and the absence of a local medical university, all within the context of an initiated pharmacogenomics program and a population-based precision medicine study known as the “Cooperative Health Research in South Tyrol” (CHRIS) study. The key strategies for addressing these challenges and integrating CHRIS study findings into a broader precision medicine development plan are discussed, including workforce development and training, investment in digital infrastructure, enhanced data management and analytic capabilities, collaboration with external academic and research institutions, education and capacity building, securing funding and resources, and promoting a patient-centered approach. This study emphasizes the potential benefits of implementing such a comprehensive development plan, including improved early detection, personal ized treatment, and prevention of chronic diseases, ultimately leading to better healthcare outcomes and overall well-being in the South Tyrolean population.

## 1. Introduction

Precision medicine, also known as personalized medicine or individualized medicine, is a medical approach that tailors diagnosis, treatment, and prevention strategies to a patient’s unique genetic, environmental, and lifestyle factors [1]. This approach may enable healthcare providers to deliver more effective and targeted interventions, improve patient outcomes, reduce adverse effects, and optimize the use of healthcare resources. In oncology, for example, it may help identify more patients who would benefit the most from “expensive” targeted therapies and recruitment into clinical trials [2].

Precision medicine may become an important health policy issue if optimized for public health [3]. Precision medicine improves the prediction and diagnosis of chronic diseases, morbidity, and mortality and can thus improve the quality of care and quality of life of people [4]. This may help to reduce time, effort, and healthcare costs [5]. Furthermore, if intentionally focused on health equity, it has the potential to address health disparities and promote health equity by considering the unique needs of diverse populations and ensuring that all individuals receive the most appropriate care [6].

Until recently, only a few European countries implemented public health policies in the field of precision medicine [7]. In Italy, recommendations for genomics in healthcare were published in 2013 in the first guidelines on genomics in healthcare [8]. The Italian National Healthcare Innovation Plan describes the need for precision medicine for educational activities in healthcare for professionals, decision makers, and citizens [9]. The Italian National Prevention Plan 2020–2025 recommends that community-based health strategies, such as promoting healthy lifestyles and personalized strategies, for example, identifying at-risk individuals or people in the early stages of a disease, should be combined and integrated [10]. This provides a mandate for the widespread development of infrastructure and prerequisites for precision medicine. Primary care and community-based interventions can reach a large proportion of the population and improve access to precision medicine services, particularly in underserved or remote communities.

In Italy, the national health service is publicly funded, similar to England’s healthcare system [11]. Within Italy, South Tyrol is an autonomous province, where the majority speak German as a second national language [12]. Given the need for health professionals to be bilingual [13,14], the current shortage of health professionals is even more serious than in the rest of Italy [15]. In addition, for analogous reasons [16,17], the healthcare sector lags behind in terms of digitalization. However, there is a need to develop modern precision medical tools for healthcare applications. No medical university or healthcare industry in the province can support this development challenge.

The aim of this article on public health is to describe the main challenges expected and faced in a development program for sustainable precision medicine in the northernmost province of Italy.

## 2. Precision Medicine and Public Health

### 2.1. Components of Precision Medicine

Understanding the genetic and molecular basis of disease has led to the development of targeted therapies that modulate specific disease pathways, resulting in improved treatment outcomes. The key components of precision medicine include genomics and genetic testing for understanding disease susceptibility and molecular profiling to assess overall health status. Pharmacogenomics tailors drug therapies, whereas advanced diagnostics improve disease detection. Data integration facilitates personalized treatment strategies, and digital health enables remote care. Ethical, legal, and social considerations ensure responsible healthcare delivery. These components work together to provide a comprehensive understanding of each individual’s unique biological and health needs, enabling healthcare providers to offer more targeted, effective, and safer treatment and prevention strategies [18].

### 2.2. Precision Medicine at the Bedside

Some of the best-studied areas where the advantages of precision medicine are evident, and public healthcare systems may be encouraged to include precision medicine in routine care programs, is the field of oncology, with targeted therapies and immunotherapies tailored to specific genetic mutations and cancer subtypes [19]. Examples include treatments for breast cancer (HER2-targeted therapies), lung cancer (EGFR and ALK inhibitors), and melanoma (BRAF inhibitors) [20].

In pharmacogenomics, the study of how individual genetic variations influence drug responses has led to personalized drug prescriptions, minimizing adverse drug reactions, and improving treatment efficacy. Existing health disparities, as they intersect pharmacogenomic research and implementation, are being identified [21].

Genetic testing and precision medicine approaches have been valuable in diagnosing and managing certain cardiovascular conditions, such as familial hypercholesterolemia, hypertrophic cardiomyopathy, long QT syndrome [22], neurodegenerative [23], and metabolic conditions [24], enabling better risk stratification, targeted therapies, and preventive measures.

Precision medicine is instrumental in diagnosing and managing rare genetic disorders such as cystic fibrosis, muscular dystrophy, and certain metabolic disorders [25].

### 2.3. Optimizing Population Health

To balance the objectives of maximizing benefits for most people and reducing health inequity within the constraints of finite resources, priority setting in precision medicine should consider high-impact diseases and conditions [26].

For population health, candidate conditions for precision medicine are diseases that are highly prevalent or have significant health consequences, such as chronic diseases (cardiovascular diseases and diabetes), neurological disorders (neurodegenerative diseases), and cancer. By focusing on these areas, precision medicine initiatives can have a broader impact on population health. Programs that aim to prevent diseases or detect them at an early stage can reduce healthcare costs and improve outcomes [18]. This may include initiatives such as genetic testing for inherited conditions [27] or screening programs for at-risk populations [28].

## 3. Challenges in the Development of Precision Medicine in South Tyrol

A development program aimed at implementing future-ready precision medicine under the conditions described in the Introduction section for South Tyrol faces several challenges. These challenges can be grouped into different areas, including language and cultural barriers, workforce shortages, lack of infrastructure, and limited digitalization. Addressing these challenges will require a holistic approach involving multiple stakeholders and a focus on capacity building, infrastructure development, digital transformation, and continuous improvement to build a sustainable and future-ready precision medicine program in the province.

### 3.1. Language, Cultural Barriers, and Workforce Shortages

Language, literacy, culture, and availability of professionals are among the determining factors in the roll-out of precision medicine at the population level [29]. Given the bilingual nature of South Tyrol, medical professionals need to be proficient in both Italian and German [13,14]. This requirement might limit the pool of available talent and make it more challenging to find qualified professionals with the necessary language skills. Addressing this challenge would require targeted language training and cultural sensitivity programs to ensure effective communication and a patient-centered approach [30,31].

Strong partnerships between healthcare system providers and healthcare providers, combined with sensible goal setting, are likely to be necessary and will help drive the implementation of personalized medicine [29]. The shortage of healthcare professionals is a pressing issue in the province. The need for bilingual and specialized medical staff as well as the lack of a local medical university could exacerbate this problem. Development programs must focus on attracting and retaining qualified professionals, including leadership, finance, policy, education, partnership, and human resource management systems [32]. When offering incentives and educational opportunities to develop the necessary skills for precision medicine, health workforce issues are to be considered interlinked functions that depend on a strong capacity for effective stewardship of health workforce policy.

### 3.2. Lack of Infrastructure and Limited Digitalization

The absence of medical faculty at South Tyrolean academic institutions may hinder research and development by increasing several barriers that may be encountered when developing precision medicine to optimize population health. Most importantly, the absence of medical faculty can limit access to expertise because of the shortage of medical experts, researchers, and educators with specialized knowledge in precision medicine. Establishing collaborations and partnerships with other institutions, both nationally and internationally, may be more difficult without the medical faculty to facilitate connections and joint initiatives. Finally, external funders may perceive the region as lacking the necessary expertise or infrastructure to successfully carry out projects.

Implementing a future-ready precision medicine program requires substantial financial and human resources. Securing funding from various sources, such as government grants, private investments, and international partnerships, is a significant challenge.

Collaborations with external research institutions, universities, and industry partners are essential for knowledge transfer and support in establishing a strong foundation for precision medicine initiatives [33].

The healthcare sector in the province lags in digitalization, which is a critical aspect of implementing precision medicine. Digitalization enables better data management, interoperability, and application of artificial intelligence and machine learning algorithms [34,35]. Precision medicine relies heavily on the collection, storage, and analysis of large volumes of patient data [36]. Ensuring data privacy and security while complying with the relevant regulations is crucial for building trust and protecting sensitive patient information. Development programs must establish robust data management practices and invest in cybersecurity measures [37,38]. Addressing this challenge will require significant investment in digital infrastructure, training for healthcare professionals, and adoption of digital health solutions.

## 4. Development Program for Precision Medicine in South Tyrol

South Tyrol has a population of approximately 550,000 people covered by the national healthcare system. Genetic diagnostics have been partially developed, and several diagnostic samples are still being sent to university laboratories outside the province. The first steps towards precision medicine have been taken in the direction of pharmacogenomics [39]. Given the relatively small population, the level of independent precision medicine that should be sought in such a development program depends on various factors. This applies most clearly to diagnostics. However, all other areas of precision medicine development are also affected.

The successful implementation of precision medicine will necessitate collaboration among healthcare providers, researchers, industry partners, and policymakers. Overcoming siloed approaches and fostering interdisciplinary collaboration are essential to drive innovation and ensure the seamless integration of precision medicine into the healthcare system.

Several priorities must be addressed to implement precision medicine in South Tyrol. Workforce development is necessary to improve skill sets and expand professional resources in the region. In addition, education and capacity building must be pursued to enhance local expertise.

Collaboration and partnerships should be sought for knowledge and resource exchange, while the technological foundation for precision medicine—digital infrastructure and data management capabilities—needs to be strengthened.

Securing funding is essential for program sustainability and interdisciplinary approaches should be integrated to encourage comprehensive and innovative solutions. These focal points will support the effective establishment of a precision medicine program in South Tyrol considering its specific limitations and needs.

### 4.1. Proposal

In general, it is important to develop a precision medicine program that is both sustainable and impactful. The program should focus on key areas where precision medicine can provide the most significant benefits to the population while also considering the available resources and infrastructure [3]. A precision medicine program in South Tyrol should deliver tangible benefits to the population while remaining sustainable and adaptable to the region’s unique needs and resources. The initial focus should be on expanding pharmacogenomics and enhancing genetic diagnostic capabilities. This can be accomplished by developing laboratory facilities, implementing advanced technologies, and training local healthcare staff. Simultaneously, it would be beneficial to foster local genomic research initiatives and expand the existing biobanks. This effort should be supplemented by creating a connected data repository integrated with electronic health records.

External collaboration should be pursued to access expertise, resources, and technology. Developing telemedicine and digital health infrastructure should also be prioritized to ensure equitable access to specialized care. Finally, the program should concentrate on prevalent health conditions in South Tyrol and promote patient engagement and education to improve health literacy and facilitate active participation in healthcare decisions.

#### 4.1.1. Pharmacogenomics

Given that the first development steps have already been taken towards pharmacogenomics [39], the first recommendation is to continue to build upon this foundation. Routine pharmacogenomic testing for medications with known gene–drug interactions should be implemented, focusing on high-impact areas such as oncology, cardiology, and psychiatry. This will help personalize treatment plans and reduce the risk of adverse drug reactions.

#### 4.1.2. Partnership with Local Research Institution and University

In South Tyrol, the causes of chronic diseases such as diabetes, cardiovascular diseases, and diseases of the nervous system, such as Parkinson’s disease, are investigated in a population-based precision medicine study by a public research institution with support from the local health service. The Institute of Biomedicine of the European Academy (EURAC) Research Centre and the South Tyrolean Health Service are working together within the framework of the population study CHRIS (Cooperative Health Research in South Tyrol) [40]. This study investigated the roles of genetic predisposition, lifestyle, and environment under such conditions [41,42,43,44,45,46,47]. The focus was on cardiovascular, neurological, and metabolic diseases. The research results serve the international scientific community and form the basis for targeted research with direct benefits for the South Tyrolean population, as they can improve the early detection and personalized treatment of diseases.

Integrating the CHRIS study into a broader development plan for precision medicine in South Tyrol could be achieved through a multifaceted approach [48] that focuses on building on the strengths of the study and leveraging its findings to improve healthcare in the region. This ongoing study should be integrated into a local precision healthcare development program by utilizing its findings for targeted interventions on prevalent chronic diseases and fostering partnerships among stakeholders. Simultaneously, a secure data-sharing platform should be created, public awareness should be enhanced, new diagnostic tools and therapies stimulated, ongoing research encouraged, and a robust framework for monitoring progress and evaluating the impact on patient outcomes and population health should be established (Table 1).

By integrating the CHRIS study into a broader development plan for precision medicine in South Tyrol, the region can capitalize on its strengths, knowledge, and resources to improve the early detection, personalized treatment, and prevention of chronic diseases. This integrated approach will contribute to better healthcare outcomes and overall well-being in the South Tyrolean population.

#### 4.1.3. Partnership with National and International Universities and Research Institutions

Collaborations with external research institutions, universities, and industry partners are essential for knowledge transfer and support in establishing a strong foundation for precision medicine initiatives [33]. As part of a cooperation agreement between the Paracelsus Medical Private University (PMU) in Salzburg and the South Tyrolean Health Service Trust, a new fund for the promotion of health research (the South Tyrolean Fund for the Promotion of Scientific Research (SFPR)) was recently established [49]. This research fund is available for joint cooperative projects with the PMU. In addition, a joint development plan for research activities is being developed in close cooperation; however, precision medicine has not yet been included.

Partnering with an international university renowned for precision medicine in a regional precision medicine development program can have significant benefits, but it may also encounter certain challenges. Collaborating with such an internationally established likely university-affiliated precision medicine center could provide access to world-class expertise, cutting-edge technologies, and resources that can help accelerate the local development of precision medicine initiatives with a focus on healthcare. Partnering with such institutions can facilitate the exchange of knowledge and best practices, helping to build capacity among healthcare professionals, researchers, and administrators in Italian provinces. Collaborating with a prestigious institution can enhance the international reputation and credibility of the South Tyrolean Precision Medicine Program, potentially attracting additional funding, resources, and talent. Such partnerships could create opportunities for joint research projects, clinical trials, and the development of innovative therapies and diagnostics tailored to the specific needs of the population. This kind of partnership can help establish connections with other leading institutions, researchers, and industry partners in the field of precision medicine, fostering a collaborative ecosystem and further advancing the program.

However, there are some notable disadvantages to consider. The geographical separation and time zone differences within a prestigious precision medicine institution could introduce logistical obstacles. Language disparities and cultural nuances may also add complexity to communication, collaboration, and knowledge exchange, potentially requiring additional resources for translation and cultural training. Discrepancies in regulatory frameworks and healthcare systems could pose difficulties in implementing research findings or practices from an international center in Italy. The risk of an imbalanced partnership exists where renowned institutions can potentially dominate decision making or resource allocation. Hence, the partnership must maintain a mutual benefit, with an emphasis on the needs of South Tyroleans. Finally, such collaborations could incur higher costs and potentially steeper fees for expertise and resources. In conclusion, partnering with a renowned international precision medicine center for a regional precision medicine development program can provide numerous benefits. However, it is essential to carefully consider the challenges and address them proactively to ensure successful collaboration.

### 4.2. Infrastructure

To develop precision medicine in South Tyrol, several requirements must be addressed to ensure the success and sustainability of the initiatives, as exemplified in the case of prostate cancer [50].

The development of robust precision medicine infrastructure necessitates investment in several key areas (Table 2). Research and diagnostic facilities must be maintained and upgraded using state-of-the-art equipment and technologies to support advanced research and diagnostic procedures. Similarly, the biobank needs enhancement for the secure collection, storage, and analysis of biological samples and health data. Furthermore, capabilities in data management and analytics, telemedicine, and digital health need to be expanded to effectively handle large volumes of health data and improve access to specialized care.

In addition to the technical infrastructure, educational resources require attention. The expansion of facilities and programs for training healthcare professionals in precision medicine is crucial. Reliable network connectivity and broadband access throughout the region are essential to ensure the efficient use of digital health tools. Furthermore, public education and awareness campaigns should be promoted to increase understanding and engagement with precision medicine initiatives. Finally, a robust monitoring and evaluation system should be established to assess the impact of the precision medicine program on patient outcomes and overall population health.

## 5. Discussion

The unique challenges faced by the South Tyrolean healthcare system include specific challenges arising from the shortage of healthcare professionals proficient in two languages for the bilingual population due to brain drain [51], and the possibly related lag in healthcare digitalization. It is important to develop digital culture and digital skills before investing in digital infrastructure and technology in moderately innovative regions such as South Tyrol [52]. Strengthening the local healthcare workforce and infrastructure is difficult. The need for investment in education, training, and capacity building for healthcare professionals must be addressed, as must the development of digital health infrastructure and telemedicine solutions to enhance access to care and expertise.

Public administration should invest mainly in digital education and partnerships, while in terms of education and training institutions, digital skills should be taught to cohorts of students and workers, particularly healthcare professionals. A development program in precision medicine for public health should create opportunities for collaboration between stakeholders as strategic and policy measures and contribute to the promotion of digital transformation processes by providing more financial resources.

Ongoing CHRIS studies will be important for informing precision medicine initiatives. The study results may be relevant for pandemic disease outcomes [46] and may contribute to the understanding of the genetic, lifestyle, and environmental factors contributing to chronic diseases in the South Tyrolean population in a relevant manner and how its findings can inform targeted interventions and prevention strategies. Therefore, it is important to integrate the CHRIS study as closely as possible in the development plan for precision medicine in South Tyrol for both local expertise and infrastructure. Establishing collaborations and partnerships with both local and external research institutions, universities, and industry partners is necessary. It is important to note that these collaborations can facilitate knowledge transfer, access to resources, and expertise to support the development of precision medicine initiatives.

Patient engagement and empowerment are general healthcare development goals. In the field of precision medicine, it is particularly important to promote patient engagement and empowerment through education on precision medicine and access to personal health data, which can help improve health literacy and encourage patients to actively participate in their healthcare decisions.

To ensure the successful implementation of precision medicine initiatives, assessing the baseline level of health literacy in the South Tyrolean population is a vital preliminary step. A comprehensive understanding of the community’s existing capacity to comprehend health-related information and navigate the healthcare system is fundamental for aligning precision medicine efforts with the community’s needs. This assessment should not only evaluate general health literacy but also focus on digital health literacy, which is crucial for the effective use of telemedicine and digital health tools. Accordingly, an evaluation of health literacy could be integrated into the development program, and strategies tailored based on the outcomes of this assessment can help foster greater patient engagement and empowerment.

The population-based genomic CHRIS study has already enabled the successful elaboration of ethical issues [53,54,55], which are critical to precision medicine, including data privacy, informed consent, and the potential for stigmatization or discrimination based on genetic information.

Securing adequate funding and allocating resources efficiently are critical for ensuring the sustainability and scalability of the precision medicine development plan. Funding costs will have to be justified by the positive results of a robust monitoring and evaluation framework to assess the impact of the precision medicine program on patient outcomes, healthcare costs, and overall population health, and how this can inform future investments and refinements to the plan.

Despite the multitude of challenges faced, South Tyrol can leverage the opportunities presented by precision medicine owing to its unique demographic composition. The region’s bilingualism and cultural diversity may serve as an asset rather than an obstacle, fostering a diverse healthcare landscape that caters to different patient backgrounds. Given the small size of South Tyrol, its population could be a manageable cohort for testing novel precision medicine interventions. Small but comprehensive interventions may serve as models for large-scale implementation elsewhere. This positioning might attract international collaborations, offering South Tyrol an advantageous platform to be at the forefront of precision medicine implementation.

Additionally, a broader societal perspective should be considered when developing precision medicine. The potential for patient empowerment and improved health literacy from precision medicine can extend beyond individual patient care. As precision medicine requires patients to be active participants in their care, the entire healthcare ecosystem might shift towards a more preventive and proactive model. This could reduce the burden on healthcare systems and improve overall public health. Moreover, the value of precision medicine extends beyond healthcare delivery and can provide benefits in areas such as the economy and education. For example, the demand for skills in data analysis, genomics, and digital health can foster workforce development and education in these areas.

## 6. Conclusions

Precision medicine has the potential to address the unique healthcare challenges in South Tyrol. Through the targeted use of CHRIS study findings and a strategic approach addressing key areas, such as workforce development, digital infrastructure, data management, and patient-centered care, existing barriers could be surmounted. This program, centered on the local context, could enable more effective disease prevention, early detection, and personalized treatment for the local population. The successful implementation of this plan relies on the commitment and collaboration of all stakeholders to realize a more efficient, sustainable, and equitable healthcare system in South Tyrol.

## Figures and Tables

**Table 1 jpm-13-00972-t001:** Integrating CHRIS study into precision healthcare.

Approach	Action
Targeted interventions and technological development	Intervention development, new tools, therapies
Stakeholder collaboration and data sharing	Partnerships, secure data platform
Public engagement and research	Awareness, ethics, ongoing research
Monitoring	Progress framework, patient outcomes

**Table 2 jpm-13-00972-t002:** Precision medicine infrastructure.

Infrastructure	Development Requirements
Research and development facilities	Maintain and upgrade existing research facilities with state-of-the-art equipment and technologies to facilitate cutting-edge research in genomics, proteomics, metabolomics, and other areas relevant to precision medicine.
Diagnostic and laboratory facilities	Upgrade and reorganize existing diagnostic and laboratory facilities to accommodate advanced genetic testing, biomarker identification, and other diagnostic procedures relevant to precision medicine
Biobanking	Upgrade the existing biobank for the collection, storage, and analysis of biological samples and health data from the South Tyrolean population
Management and analytics capabilities	Develop secure data management systems and analytics platforms to store, analyze, and share large volumes of health data, including genomic, clinical, and lifestyle information
Telemedicine and digital health infrastructure	Invest in telemedicine solutions and digital health tools to enhance access to specialized care and expertise, especially for patients in remote areas or with limited transportation options
Health information exchange and electronic health records	Implement a health information exchange system that allows for the seamless sharing of electronic health records among healthcare providers and institutions, enabling more efficient and personalized care
Education and training facilities	Expand facilities and programs dedicated to training healthcare professionals in precision medicine, including specialized courses, workshops, and seminars
Network connectivity and broadband access	Ensure reliable network connectivity and broadband access throughout the region, enabling the efficient use of telemedicine, digital health tools, and data management systems
Public awareness and education infrastructure	Expand infrastructure for public education and awareness campaigns, such as community centers or digital platforms, to promote understanding and engagement with precision medicine initiatives
Monitoring and evaluation systems	Establish robust monitoring and evaluation systems to assess the impact of the precision medicine program on patient outcomes, healthcare costs, and overall population health, which can inform future investments and adjustments

## Data Availability

No new data were created.

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
