# Peer review of "Advancing Precision Medicine in South Tyrol, Italy: A Public Health Development Proposal for a Bilingual, Autonomous Province"

_jpm, 2023, doi:10.3390/jpm13060972_

Round 1
Reviewer 1 Report
1- Table 1 is not necessary. Embed the information in the text
2- Avoid listing the information in the text. Explain them in a narrative tone
3- Table 2 is based on the other people’s work. Please elaborate it in the text instead of listing them in a large table
4- Table 3 seems to have the same issue, instead of listing, try to combine the information and make a comprehensive table about the issue
5- 4.1.3 also has the same issue
6- Numbering of the tables are wrong, table 1 on page 8 should be numbered after the other tables,
7- Please summarize the table on page 8
8- Discussion seems to be short and not sufficient
9- Conclusion is broader than the main conclusion of this paper
Reviewer 2 Report
I would like to congratulate the author for the article presented, not only for the way he does it, but also for the relevance of what he proposes. In fact the need to bring the practice of health professionals closer to the community is fundamental, even more so in a community with the characteristics of the one presented here.
The only suggestion I leave to the author is related to the, in my opinion, little relevance given to the issue of assessing the level of health literacy of the population. The issue of literacy is mentioned several times (and also refers health literacy) but never proposes to analyse this fundamental aspect for the adherence of the community to precision medicine. Aspects such as those mentioned in Table 2 "Develop telemedicine and digital health infra-structure" can only be achieved if there is effective health literacy and digital health literacy by all stakeholders. At the end of the same table it states that "Increase patient engagement and education" can "help improve health literacy". I agree, but we need to establish the baseline of this level of health literacy. So it seems to me that this aspect should also be included in the aspects mentioned in lines 311 to 315. Although very interesting and relevant, I suggest adding some information regarding the assessment of the health literacy level of the population.
Round 2
Reviewer 1 Report
Thank you for addressing the comments and providing answers. I appreciate your assistance. I don't have any further questions at the moment.